# INSYDE-BE: Adaptation of the INSYDE model to the Walloon Region (Belgium)

Anna Rita Scorzini[1], Benjamin Dewals[2], Daniela Rodriguez Castro[2,3], Pierre Archambeau[2], Daniela Molinari[3]

[1] Department of Civil, Environmental and Architectural Engineering, University of L'Aquila, L'Aquila, 67100, Italy
[2] Hydraulics in Environmental and Civil Engineering (HECE), University of Liège, Liège, 4000, Belgium
[3] Department of Civil and Environmental Engineering, Politecnico di Milano, Milano, 20133, Italy

*Correspondence to*: Anna Rita Scorzini (annarita.scorzini@univaq.it)

**Abstract.** The spatial transfer of flood damage models among regions and countries is a challenging but unavoidable approach, for performing flood risk assessments in data and model scarce regions. In these cases, similarities and differences between the contexts of application should be considered to obtain reliable damage estimations and, in some cases, the adaptation of the original model to the new conditions is required. This study exemplifies a replicable procedure for the adaptation to the Belgian context of a multi-variable, synthetic flood damage model for the residential sector originally developed for Italy (INSYDE). The study illustrates necessary amendments in model assumptions, especially regarding input default values for the hazard and building parameters and damage functions describing the modelled damage mechanisms.

## 1 Introduction

With the shift from hazard control to risk mitigation policies, flood damage assessment has gained increased importance in the last decades as a key information tool for effective flood risk management. Indeed, knowledge of flood damage is crucial both in the emergency (to identify priorities of intervention and to support the compensation of damage by private and public bodies) and in the peace time (to identify areas at higher risk and to evaluate benefits of flood mitigation strategies).

To date, several flood damage models have been developed by many authors for various exposed assets, with the residential sector representing the most investigated one (Merz et al., 2010; Gerl et al., 2016). However, flood damage modelling is still often hampered by the paucity of sufficient and high-quality data for model calibration and/or validation. When this occurs, the spatial transfer of models developed in different regions than the one under investigation could be a choice. However, this must be done cautiously, considering similarities and differences between the compared contexts and, if required, by adapting the original model to the new conditions (Cammerer et al., 2013; Saint-Geours et al., 2014; Amadio et al., 2019; Molinari et al., 2020). In this framework, the present study discusses the transferability of the Italian model INSYDE to the Belgian context and specifically to the Walloon region.

Damage models can be distinguished in two types, namely empirical and synthetic, depending on the approach used for their development: empirical models are based on loss data observed in actual flood events, while synthetic approaches use expert information on damage mechanisms collected via "what-if" questions (Merz et al., 2010; Sairam et al., 2020). The development of empirical models is currently limited by the endemic paucity of ex-post damage data, discussed above; this is especially true in Belgium where the co-existence of a public and a private system for damage compensation leads to the availability of partial and unrepresentative damage data (Doppagne, 2020). Moreover, the often implicit formulation of empirical models ("black-box models") is usually a limiting factor for their spatial transferability, given the possible differences in hazard and vulnerability features characterizing the original context of derivation of the model and the new area of application. On the other hand, synthetic models can be, in principle, derived in any region, as long as sufficient expertise on damage phenomena under investigation exists; still, their development may be a long and challenging process, depending on the level of detail reached by the model and the number of explicative variables taken into account. Indeed, flood impacts depend on the interaction between several flood characteristics and vulnerability parameters, such as water depth, flow velocity, inundation duration, floodwater contamination and sediment load, as well as building material, construction type, building age and quality (Penning-Rowsell et al., 2005; Thieken et al., 2005; Dottori et al., 2016; Mohor et al., 2020). Accordingly, multi-variable models have been recently proved to increase the accuracy of damage estimations compared to traditional, simple stage-damage curves (Schröter et al., 2014, 2018; Dottori et al., 2016; Wagenaar et al., 2018; Amadio et al., 2019; Sairam et al., 2020). Moreover, modeling of damage mechanisms at the base of synthetic models make them better suited to be transferred from one region to another, of course, after a careful check of the comparability between the original and the new physical and economic contexts of implementation and, if required, a consequent adjustment of model inputs and assumptions (Lüdtke et al., 2019; Scorzini et al., 2021).

INSYDE is an example of a synthetic, multi-variable flood damage model for the residential sector, originally developed and validated for Italy (Dottori et al., 2016, Amadio et al., 2019; Molinari et al., 2020). Despite its complexity, the clarity in the assumptions and the flexibility for the adaptation of the input parameters and the mathematical functions describing the damage mechanisms support its transferability to regions different from the original one.

In this paper, the steps needed for the adaptation of INSYDE to the Belgian context, and specifically to the Walloon region, are first discussed (section 2); then, results of the adaptation process are presented, in terms of new or adjusted input variables and functions, leading to the INSYDE-BE model (section 3). A critical discussion of the usability of the newly developed model and final remarks close the paper.

## 2 Data and methods

### 2.1 INSYDE

INSYDE is a synthetic, multi-variable flood damage model, released as an open-source R script, which estimates economic damages for residential buildings using expert-based mathematical functions (Dottori et al., 2016). The total damage per

building ($D$) is expressed as the sum of the costs for repairing (or removing and replacing) the various affected building components ($C_i$), which are further divided into sub-components ($C_{ij}$), as follows:

$$D = \sum_{i=1}^{n} C_i = \sum_{i=1}^{n} \sum_{j=1}^{m_i} C_{ij} \tag{1}$$

The damage cost to each subcomponent $C_{ij}$ is expressed as a function of the damage extension ($ext_{ij}$) in physical terms (e.g., m$^2$ of damaged pavement), multiplied by the unitary price ($up_{ij}$) of a specific activity regarding a damaged building component (e.g., cost of pavement replacement per m$^2$), and an additional factor ($r_{ds}$) depending on the modelled damage mechanism, i.e., deterministic or probabilistic (probability of damage occurrence to the considered component as a function of a certain hazard characteristic intensity):

$$C_{ij} = ext_{ij} \cdot up_{ij} \cdot r_{ds} = f \text{ (hazard and building features, unit prices)} \tag{2}$$

In particular, $ext_{ij}$ is a function of several (23) damage explicative variables, related both to the hazard (6: external and internal water depth, flow velocity, inundation duration, sediment load and presence of contaminants) and to the characteristics of the affected building (17, including, among others: geometric (e.g., footprint area, internal and external perimeters) and qualitative features (e.g., building type and quality, level of maintenance)). In case of gaps in input data availability (or requiring extensive data collection), the model proposes default values for each parameter. Moreover, the model proved to be adaptable to the actual available knowledge of the flood event and building characteristics, with the possibility of downscaling information available at meso-scale (Molinari and Scorzini, 2017).

## 2.2 Study area – Walloon Region

The Walloon Region corresponds to the southern half of Belgium (Figure 1) and has a population of 3.6 million inhabitants. It covers an area of nearly 17,000 km$^2$ and is divided into five provinces (Hainaut, Liège, Luxembourg, Namur and Walloon Brabant), which correspond to level 2 in the European nomenclature of territorial units for statistics (NUTS). Four international river basins cover parts of the Walloon Region: mostly Meuse (73 % of the territory), Scheldt (22 %), as well as tiny fractions of the Rhine and Seine basins. The mean annual precipitation ranges between 700 and 1,400 mm, and snowmelt may influence flood discharges in some parts of the Meuse basin. As a result of historical developments, particularly industrialization, densely urbanized areas are to a great extent concentrated along the main rivers (Poussard et al., 2021).

Existing flood hazard maps cover the whole region. For the main rivers, they were generated based on two-dimensional hydraulic modelling, combined with laser altimetry and sonar bathymetry data. Three scenarios were considered in the preparation of these hazard maps: 25-year, 50-year and 100-year floods. These computations provide water depth and flow velocity in the main riverbed and throughout the floodplains, with a grid spacing ranging between 2 m and 5 m (Erpicum et al., 2010). These hydraulic data were used in several previous research (Bruwier et al., 2015; Detrembleur et al., 2015; Mustafa et al., 2018).

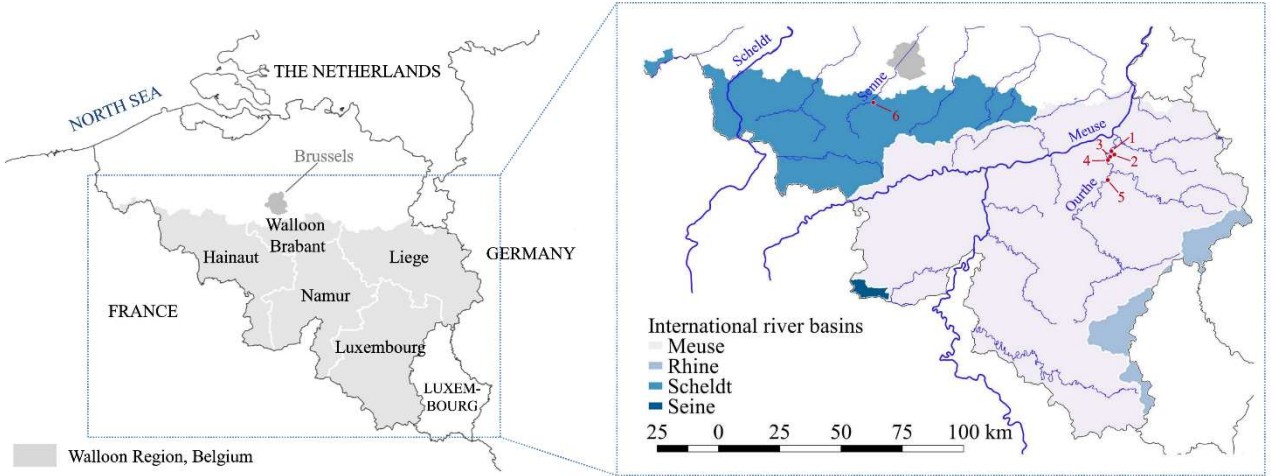

**Figure 1: Location of the case study area (Walloon region) in Belgium, with indication of the names of the provinces. The numbers in the map on the right refer to the municipalities where field surveys were conducted: Tilff (1), Méry (2), Hony (3), Esneux (4), Comblain-la-Tour (5) and Tubize (6).**

## 2.3 Adaptation of INSYDE

A new methodology has been developed to adapt the INSYDE model to the Walloon region. As shown in Figure 2, the procedure consists in two main steps. The first one aims at analyzing the characteristics of the typical flood events and the features of residential dwellings in the new region, to check the transferability or the need for modification of model inputs (and related default values), and assumptions on damage mechanisms implemented in INSYDE. Regarding the hazard

components, this step consists in the analysis of hydrological and hydraulic records and information about historical flood events, of existing flood hazard and risk maps as well as a review of past and ongoing projects on flood damage estimation. For the characterization of the housing stock, the analysis involves literature review of papers and statistical reports concerning the building spatial structure, urban plans or future tendency of renovation and construction in the region of interest. The step of data collection also includes virtual and/or field surveys, aimed at obtaining specific information (e.g.,

typical features of building interiors) not available from a standard desk-based data retrieving process.

Once the required information for model adaptation is collected, statistical analysis of the data allows defining characteristic values for both hazard and building parameters, to be set as default values in the model. The adaptation also requires the change of the unit prices for the operations of removal and replacement of building components considered in the model, with values representative for the new implementation context.

The second step of model adaptation also includes the adjustment of the damage functions based on the results of the previous steps to represent more accurately the flood damages typically experienced in the implementation context. These adjustments may include amendments both in the extension of the damaged components ($ext_{ij}$ in Eq.2) and the thresholds for damage occurrence (i.e., fragility functions for the various components, $r_{ds}$, in Eq. 2) or the creation of new specific functions.

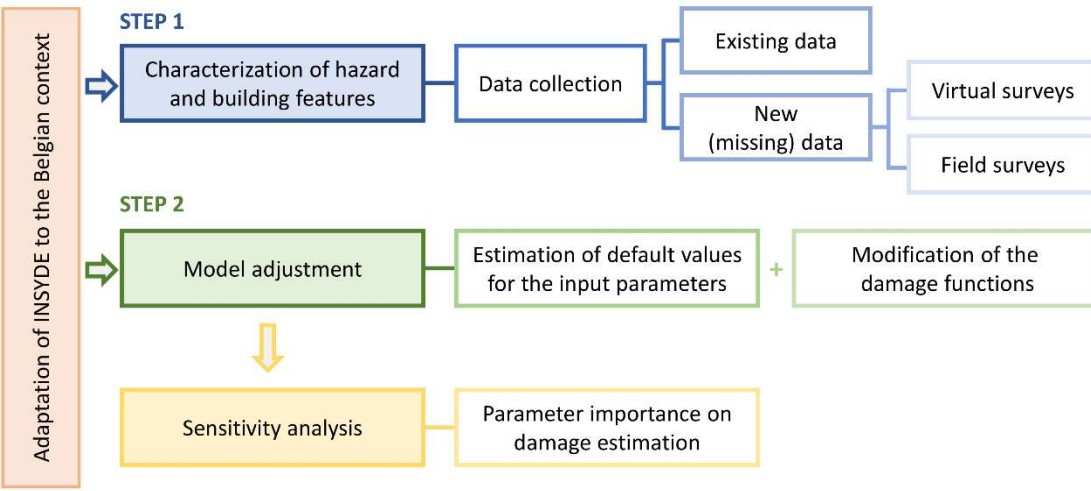

**Figure 2: Flowchart for the adaption of INSYDE.**

## 2.4 Sensitivity analysis for the new adapted model

Once the model is adapted to the new implementation region, a sensitivity analysis is recommended to assess the influence of the various hazard and building variables on the total damage estimation. This analysis provides valuable information for practical application of the model, by pointing at either mandatory, or (possibly) redundant input parameters that may be set at their default values without significantly affecting the results. For the case under investigation, the sensitivity analysis was conducted by multiple damage simulations in which single input variables were varied one at the time, while keeping the others constant. The results were analyzed in terms of a sensitivity score, calculated as the ratio of the difference $\Delta D_i^{\pm}$ in the damage obtained for the two possible extreme values of a variable ($x_i^{+}$, $x_i^{-}$) to the estimated damage $D^0$ when the variable is set at its baseline (default) value ($x_i^0$), as follows:

$$\frac{\Delta D_i^{\pm}}{D^0} = \frac{D(x_i^{\pm}, x_j^0) - D(x_i^0, x_j^0)}{D(x_i^0, x_j^0)} \tag{3}$$

where $x_j^0$ denotes the other variables that are kept constant to their default values during the tests. The results of the hazard and building data statistics were considered for the definition of the baseline scenarios as well as of the extreme bounds for the tested parameters. In detail, two different sets of hazard scenarios were identified as baseline for the sensitivity analysis of the hazard and buildings features, respectively. With respect to hazard parameters, baseline scenarios were chosen in order to gain insights on the variability of damage estimations for the typical flooding events in the Walloon region. As regards building features, baseline hazard scenarios were instead defined to appreciate differences in damage estimations that are linked to the triggering of the various damage mechanisms implemented in the model (i.e., identifying the main threshold values for the hazard parameters causing the occurrence of damage to specific building components).

**3 Results and discussion**

**3.1 Adaptation of INSYDE**

An analysis of collected data was performed to identify representative values of the model parameters for the Walloon region and to establish relationships among them, with the double aim of assigning default values (Table 1 for hazard variables and Table 2 for building variables) and modifying, if required, the damage functions to the new context of application (Table 3). The results of this analysis are reported hereinafter.

**3.1.1 Hazard features**

*Water depth*

The variability of water depth was assessed by considering the 25-, 50- and 100-years return period flood hazard maps available for 35 rivers in the Walloon region, consisting in raster files with a spatial resolution of 5 m obtained from 2D hydraulic modelling (Detrembleur et al., 2015; Erpicum et al., 2010; Mustafa et al., 2018). The distributions of water depths are shown in Figure 3 by means of boxplots, where the rivers have been classified according to the two main districts of the region, Scheldt and Meuse (Figure 1). Median flood depths range between 0.3 and 0.65 m in the Scheldt district, with limited variability among the various scenarios, expect for a 100-year flood in the Scheldt river, showing larger values, with a median flood depth of 1.7 m.

For the Ourthe river, which flows in the area where the field surveys were carried out in the province of Liège (see section 3.1.2), the median depths vary between 0.48 and 0.62 m for the 25- and 100-year floods, respectively, with maximum values up to 2 m. These values were confirmed by the field surveys, where interviewed inhabitants of Tilff and Esneux indicated observed water depths for the 1993 event, ranging from 0.4 to 1.6 m. According to this analysis, the range of water depths considered in INSYDE-BE was considered not to exceed 3 m.

*Flow velocity*

Flow velocity is another important factor for flood damage estimation, especially when it reaches medium to high values that may cause damages to secondary elements, such as doors and windows, or to the structural components of the building. The hazard maps discussed in the previous section also provided information on this parameter, enabling the characterization of typical flow velocities for floods occurring in the region, by analyzing the empirical distributions obtained from the processing of the raster files. An example of the results is reported in Figure 4 for three flood scenarios in the Ourthe river. While indicating small variability between the different scenarios and a low chance for the occurrence of significant structural damages (given the indicated low velocity values, e.g., 90[th] percentile = 1.5 m/s (Clausen and Clark, 1990)), Figure 4 suggests that 0.5 m/s (corresponding to the median value observed in the distributions) can be considered a proper default value to be assigned to this parameter in INSYDE- BE.

**Table 1: Hazard parameters included in INSYDE-BE: default values, with indication of the type of adaptation required with respect to the original model and information on the data source(s) which supported the process of adaptation.**

| Var. | Description | Unit of meas. | Range of values | Default values | Adaptation with respect to the original model | Data source |
|---|---|---|---|---|---|---|
| $h_e$ | Water depth outside the building | m | $\geq 0$ | [0; 5] incremental step: 0.01m | - | Computed water depth in the flooded area, for return periods of 25, 50 and 100 years, for the 35 Walloon river catchments |
| $h$ | Water depth inside the building | m | [0; IH] | $h = h_e - GL$ | - | - |
| $v$ | Maximum velocity of the water perpendicular to the building | $ms^{-1}$ | $\geq 0$ | 0.5 | - | Computed water velocity in the flooded area of the river Ourthe, for discharges of 25, 50 and 100 years of return period |
| $d$ | Flood duration: persistence of water inside the building | hours | $> 0$ | 34 | Change in the default value | Field surveys of past events for different basins in the Walloon region (University of Liège) / Field surveys |
| $s$ | Sediment load | % on the water volume | [0;1] | 0.05 | - | Field surveys |
| $q$ | Water quality: presence of pollutants | - | 0 = no 1 = yes | 1 | - | Field surveys |

**Table 2: Building parameters included in INSYDE-BE: default values, with indication of the type of adaptation required with respect to the original model and information on the data source(s) which supported the process of adaptation.**

| Var. | Description | Unit of meas. | Range of values | Default values | Adaptation with respect to the original model | Data source |
|---|---|---|---|---|---|---|
| FA | Footprint area | $m^2$ | $> 0$ | 110 (detached) 75 (semi-detached) 75 (attached) 95 (apartment) | Change in the default value | Statistical data Virtual and field surveys |
| IA | Internal area | $m^2$ | $> 0$ | $0.9 \cdot FA$ | - | Virtual and field surveys |
| BA | Basement area | $m^2$ | $\geq 0$ | $0.5 \cdot FA$ | - | Virtual and field surveys |
| EP | External perimeter | m | $> 0$ | $4\sqrt{FA}$ (detached/apartment) $\sqrt{2FA}$ (attached-centre) $2\sqrt{2FA}$ (attached-corner) $3\sqrt{FA}$ (semi-detached) | Change in the default value | Synthetic analysis |
| IP | Internal perimeter | m | $> 0$ | $0.64 \cdot FA + 17.02$ (detached/apartment) $0.42 \cdot FA + 27.29$ (attached) $0.56 \cdot FA + 12.9$ (semi-detached) | Change in the default value | Virtual and field surveys Synthetic analysis |
| NF | Number of floors | - | $\geq 1$ | 2 | - | Statistical data Virtual and field surveys |
| IH | Interfloor height | m | $> 0$ | 3.5 | - | Virtual and field surveys |
| BH | Basement height | m | $> 0$ | 2.5 | - | Virtual and field surveys |
| GL | Ground floor level | m | $\geq 0$ | 0.2 (detached, attached, semi-detached) 0.1 (apartment) | Change in the default value | Virtual and field surveys |
| BL | Basement level | m | $\leq 0$ | $-GL - BH - 0.3$ | - | Virtual and field surveys |
| BT | Building type | - | 1 = detached 2 = semi-detached 3 = attached 4 = apartment | 3 | Additional building type included and Change in the default value | Statistical data and grey literature Virtual and field surveys |

| BS | Building structure | - | 1 = reinforced concrete 2 = masonry | 2 | - | Statistical data and grey literature Virtual and field surveys |
|---|---|---|---|---|---|---|
| FL | Finishing level | - | 0.8 = low 1 = medium 1.2 = high | 1 | Change in the default value | Grey literature Virtual and field surveys |
| LM | Level of maintenance | - | 0.9 =low 1= medium 1.1= high | 1 | Change in the default value | Grey literature Virtual and field surveys |
| YY | Year of construction | - | $\geq 0$ | 1940 | Change in the default value | Statistical data Virtual and field surveys |
| PD | Heating system distribution | - | 1= centralized 2= distributed | 1 (if YY $\leq$ 1990) 2 (otherwise) | - | Grey literature Virtual and field surveys |
| PB | Building position (if BT = 3) | - | 1 = corner 2= center 3= else | 2 | New variable included in the model | Virtual and field surveys |
| EFM | Exterior finishing material | - | 1 = plaster 2= stone 3= masonry 4 = stone & bricks | 3 | New variable included in the model | Grey literature Virtual and field surveys |

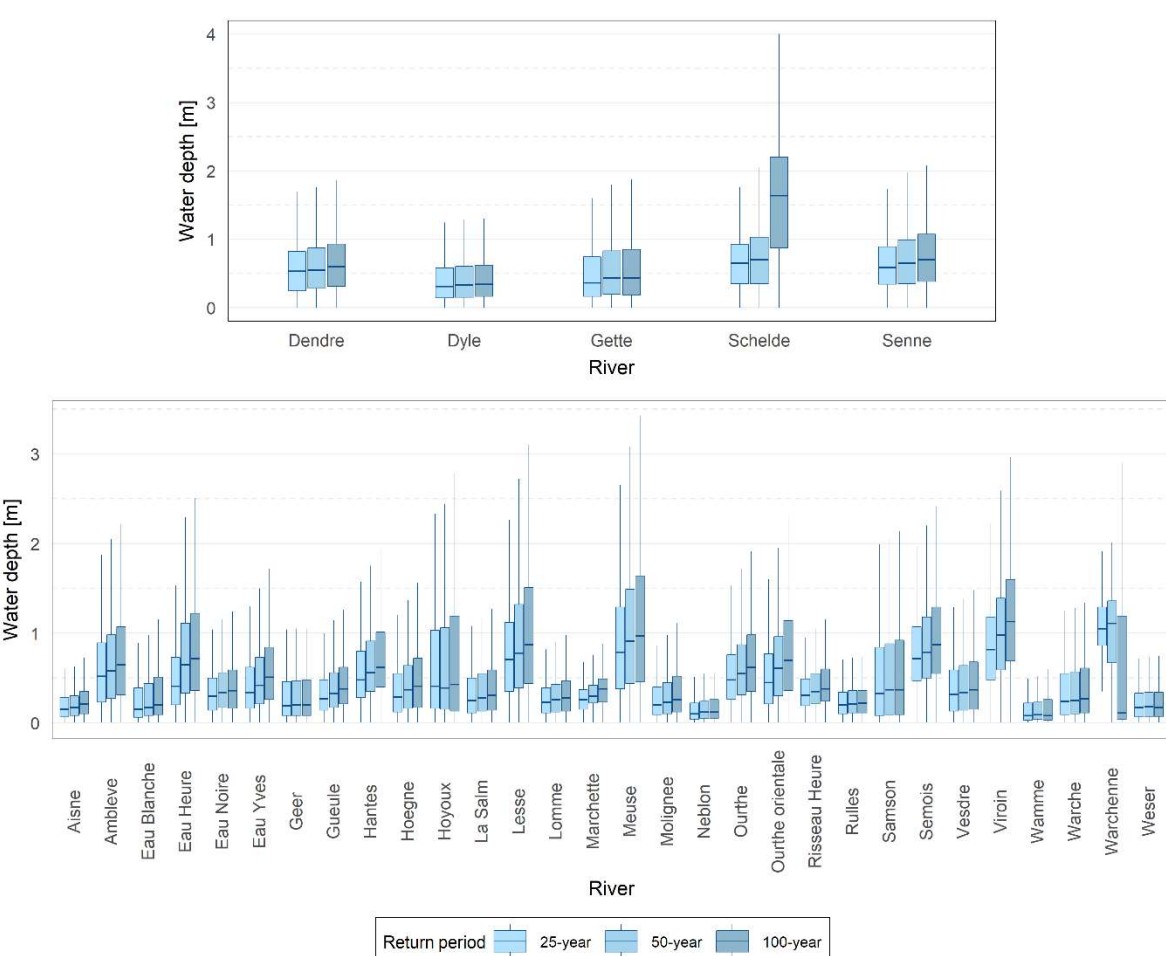

**Figure 3: Expected water depths for different return period floods in the two main river districts in the Walloon Region: Scheldt (top) and Meuse district (bottom).**

*Flood duration*

The characterization of the flood duration was based on the analysis of a database developed in the framework of a project led by the University of Liège (Petit et al., 2005), which reports different information about past flood events (including flood duration) gathered through field surveys and interviews in the Walloon region. The observed durations for all the surveyed river basins are summarized in Figure 5, which indicates 34 hours (i.e., the average of all the reported durations) as

a representative default value to be assumed in INSYDE-BE for the flood duration. These data were also cross-checked with the information collected during the field surveys along the Ourthe river, performed in the context of the present study (see section 3.1.1.2), where the interviewees reported durations ranging between 2 and 3 days.

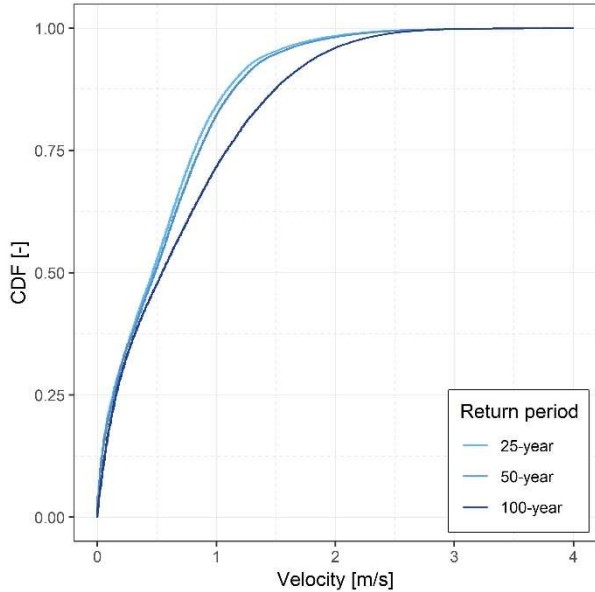

**Figure 4: Cumulative distribution functions for the flow velocity in the Ourthe river for different return period flood scenarios.**

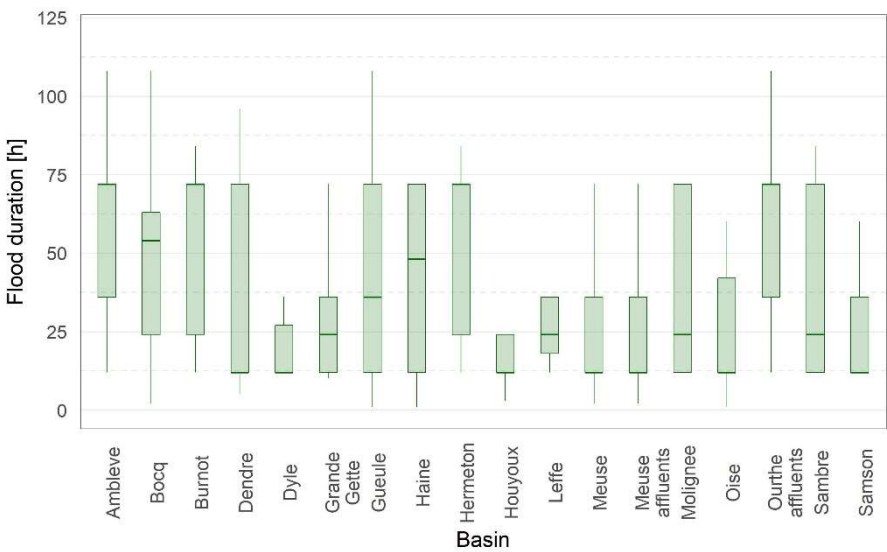

**Figure 5: Summary of reported flood duration for past events in the river basins of the Walloon region (elaborated from data derived from Petit et al., 2005).**

*Sediment load and water quality*

From the desk-based analysis, no useful data were found for a clear description of the typical characteristics of the flood events occurring in the Walloon region in terms of sediment load and water quality. Consequently, the default values for these parameters were assumed based on the qualitative information gathered through the field surveys (see section 3.1.1.2). Indeed, more than 60% of the people interviewed along the Ourthe and Senne rivers described the flood water as "dirty or

muddy". Therefore, the presence of sediments was considered as necessary to be included in the model, with a default value

equal to 5% of the water volume, as in the original INSYDE model developed for Italy. The water quality is instead a binary variable, referring to the possible presence of pollutants. Also in this case, the field surveys supported the importance of including this variable in the model, with a default value set to 1 (i.e., presence of pollutants) given that many inhabitants reported memories of gasoline smell in the floodwater.

### 3.1.2 Building features

The identification of the typical main features of residential buildings in the Walloon region, was mostly obtained from the information collected through the virtual and field surveys, after a preliminary desk-based analysis of national and/or regional databases, as well as of specific literature on the characterization of the spatial structure and the housing quality in the Walloon region (Carlier et al., 2007; Vanneste et al., 2007, 2008; Singh et al., 2013; Anfrie et al., 2014). The virtual surveys consisted in the screening of real estate websites in Belgium (e.g., immoweb.be and homelog.be). The systematic

analysis of building photos and description of the property given by the agent/owner allowed the collection of useful information to identify typical Belgian standards for certain ultra micro-scale building parameters strictly required in INSYDE (e.g., typical windows size) or playing an important role in the occurrence of specific flood damage mechanisms (e.g., approximate height of lower and middle sockets, for determining the water depth threshold for damage occurrence to the electrical system). About 230 buildings located in urban and rural areas within the five provinces of the Walloon region

were surveyed, extracting all the data shown in the Repository (doi: 10.17632/7ckzzz3xz5.1 – "Summary_virtual_surveys_Wallonia"). Field surveys aimed at corroborating the information obtained from the desk-based analysis and the virtual surveys, but, more importantly, at gaining knowledge on flood damage mechanisms, by interviewing residents who experienced past flood events. A total of 32 interviews were carried out in January-February 2020 in flood prone areas in the provinces of Liège (along the Ourthe river, specifically in the localities of Mery, Hony, Tilff, Esneux and

Comblain la Tour) and Walloon Brabant (Tubize, along the river Senne), which were affected by multiple flood events over the last 30 years. The questionnaire developed within the RISPOSTA project (Ballio et al., 2018) was used to collect the information related to the hazard and building features, to the type(s) of damages suffered in the past floods, as well as to the mitigation measures implemented by the residents (if any). The summary of the information collected in the field surveys is reported in the Repository (doi: 10.17632/7ckzzz3xz5.1 – "Field_surveys_Walloon_Region_2020").

*Micro-scale building features*

According to the literature review and the analysis of statistical data, residential buildings in Belgium can be classified into four categories: attached (building joined to other houses in one or both sides, typically constructed along the streets), detached and semi-detached houses, and apartments. Single-family buildings are the most common type, while apartments represent no more than 21% of the total housing stock. Attached and semi-detached houses are the most frequent categories,

with a share of 27% and 23% of the stock, respectively. There is also a small share of trailer homes serving as permanent dwellings, but these tend to disappear in line with future urban and flood risk management plans (Carlier et al., 2007;

Vanneste et al., 2007, 2008; Opdebeeck and De Herde, 2014). Therefore, in contrast with Italy, a vast majority of the Belgian building stock is made of single houses. Besides this classification, a distinction can also be done according to the architectonic development of the building stock throughout the years (*maison modeste, maison moyenne, maison de maître, maison appartements* (Singh et al., 2013)), with more than 50% of it built before 1945 (Vanneste et al., 2007, 2008; STATBEL, https://census2011.be), especially for attached and semi-detached buildings, as evident from the surveys. As in Italy, typical materials of constructions are masonry or reinforced concrete, with the former being the predominant type (Singh et al., 2013).

Concerning the geometric features of the buildings, the only parameter available at the regional level is the footprint area, downloadable as a vector file from the geoportal of the Walloon region (PICC data, Service public de Wallonie: https://geoportail.wallonie.be/). The processing of these data allowed us to calculate the median footprint area for residential buildings in each province (Figure 6), with Hainaut and Liège characterized by smaller buildings (57 and 68 m$^2$, respectively), followed by Walloon Brabant and Namur (around 85 m$^2$), and Luxembourg being the province with larger buildings (103 m$^2$). Figure 6 also shows the presence of residential buildings larger than 1000 m$^2$, which, however, represent less than 0.5% of the total in each province; moreover, as revealed by a detailed analysis of this sample, these cases are usually associated with large apartment blocks, which are not the most representative nor frequent building type in Belgium. Information on the footprint area was also retrieved from virtual surveys, in particular with respect to the dimension of a single housing unit composing the building. From these data, a clear relationship between the size of the housing unit and the building type was recognized, leading to the calculation of a median size of a housing unit for each building category (Figure 7), following adopted as default values for the footprint area as a function of the building type (Table 2).

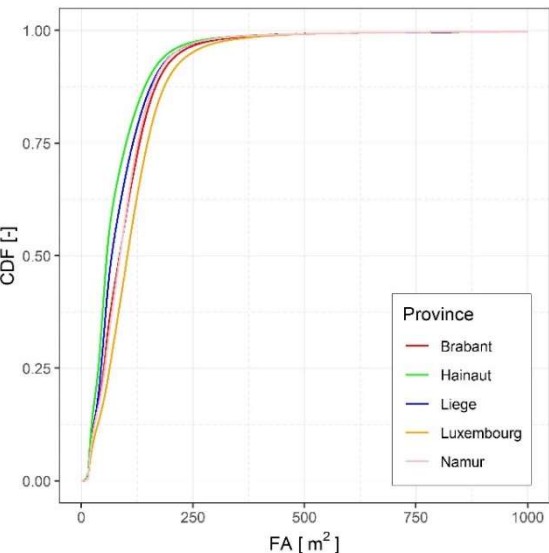

**Figure 6: Empirical cumulative distribution functions for the footprint area in the five provinces of the Walloon region (based on PICC data). Median values: Brabant: 85.4 m$^2$; Hainaut: 56.8 m$^2$; Liege: 67.7 m$^2$; Luxembourg: 102.9 m$^2$; Namur: 86.3 m$^2$.**

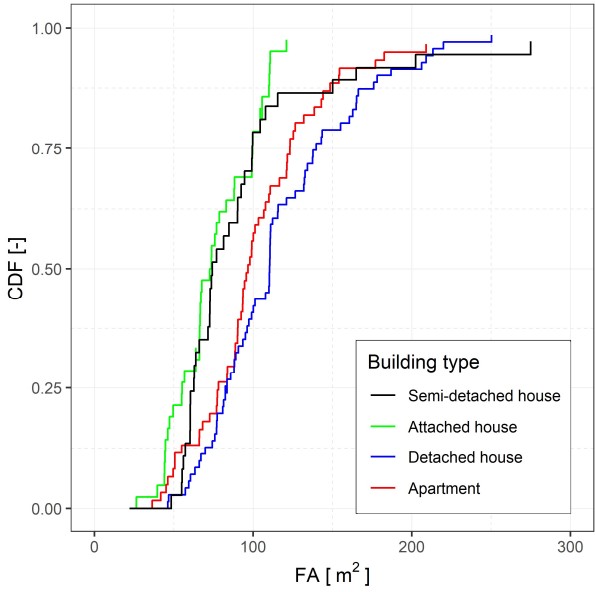

**Figure 7: Empirical cumulative distribution functions for the footprint area as a function of the building type (based on data retrieved from the surveys). Median values for each category are reported in Table 2.**

Differently than in the original INSYDE model, in INSYDE-BE the variable FA refers then to the single housing unit and not the whole building. This choice is in accordance with the limitation of the original model as regards the definition of FA highlighted by Galliani et al. (2020).

*Ultra micro-scale building features*

Variables characterizing the building at the ultra micro-scale were defined based on the information collected through the virtual and field surveys. Hereafter, we report examples of the attribution of default values to such variables, while all the implemented values are summarized in Table 2.

In the surveys, it was observed that a common feature of residential buildings in the Walloon region is to have the ground floor elevated from the street level to a certain extent (generally a few steps). This ground floor level (GL) was found to depend on the building type. Indeed, by analyzing the database of the virtual surveys, an average elevation of 0.29 m (and a median of 0.15 m) was found for semi-detached and detached buildings, while slightly lower values were detected for apartment buildings, characterized by an average of 0.2 m (and a median of 0.10 m). For this reason, differently from INSYDE-Italy, for the Walloon region two default values were specified, being 0.1 m for apartment buildings and 0.2 m for all other building types.

From the data derived from the virtual surveys, it was also possible to establish an empirical relationship between the footprint area and the basement area, i.e., $BA = n \cdot FA$, with observed mean values of $n$ equal to 0.54 (and median of 0.43),

which allowed us to assume as default value for BA the following equation $BA = 0.5 \cdot FA$.

Synthetic or mixed empirical-synthetic approaches were followed for the other geometric parameters of the building, such as the external and internal perimeter. For example, a relationship for the external perimeter (EP) with the footprint area (FA) of the housing unit was synthetically derived (Table 2), for each building type, by considering typical external and internal layouts, as represented in Figure 8.

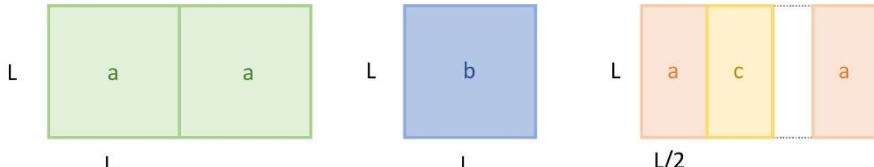

**Figure 8: Building layouts implemented in the synthetic derivation of the relationship between the external perimeter and footprint area as a function of the building type. Three different typologies of housing units can be distinguished a) housing units located in the corner of a building; b) isolated housing building; c) housing units located in the center of a building.**

Differently from the original model, the analysis led to the necessity of distinguishing between housing units located in the

280 corner and in the center of attached buildings (Figure 8), as they are characterized by different EP-FA relationships, and then to the creation of a new variable identified as "building position" (PB). A combination of an empirical and synthetic analysis was instead performed to derive a relationship between the internal (IP) and external (EP) perimeters. We combined empirical data on IP taken from the virtual surveys, with those generated by the application of a synthetic approach, consisting in the creation of hypothetical building layouts, by assuming realistic internal distributions of the rooms,

depending on building size and type. By regression analysis, it was found that IP correlated better with FA than EP (details in the Supplement). The relationships shown in Table 2 were implemented in the model.

Another example of amendment to the original INSYDE model is represented by the introduction of a new parameter accounting for the type of exterior finishing material (EFM). Indeed, from the surveys it was recognized that in Belgium the exterior of the buildings is typically mixed and composed of two materials, one located in the lower part of the building wall

(i.e., from the ground to an average height of 0.6 m, usually made of stone) and another one (generally masonry) in the upper part. However, other external finishing materials are also present and cannot be ignored within the model, leading to four possible choices for EFM (Table 2).

### 3.1.3 Update of the unit prices

In order to update the unit prices corresponding to the fixing of the damage to the various subcomponents identified in

INSYDE-BE, the required materials, tools or equipment and the workforce for each restoration activity were analyzed. The main source for estimating these costs was the *"Bordereau des Prix Unitaires 2020"*, compiled by ABEX (Association

Belge des Experts) and UPA (Union royale Professionnelle d'Architectes). The final estimation of the unitary prices for each damage subcomponent considered in the model is provided in Table S1 in the Supplement.

### 3.1.4 Modification of the damage functions

Based on the characterization of the typical flood events in the region as well as of the quality and features of exposed dwellings, all the damage functions considered in the original INSYDE model were analyzed and their validity for the new context was checked. When necessary, amendments were proposed to reasonably represent the damage scenarios in the Walloon region. This section describes some examples of adaptation, while the whole set of damage functions implemented in INSYDE-BE is reported in the Supplement and summarized in Table 3. As in the case of the definition of the default

values, this operation was supported by the information derived from the virtual and field surveys.

A first example is given by the damage subcomponent "Dehumidification", which depends on the flood duration, the perimeter and the area of the flooded floors. During the interviews with people who experienced flood events, it emerged that this activity is usually performed only for moderately long-lasting flooding. This information was then used to modify the related fragility function, by shifting the thresholds for damage occurrence, now starting at 24 hours and reaching a 100%

probability at 48 hours.

Other components that required a substantial change were the ones related to the flooring system, due to the different features of the typical construction types used in Italy and in Belgium. Indeed, the Italian flooring system is usually characterized by the presence of the screed, i.e., a concrete layer laid on the top of the slab, over which the ceramic or parquet floor is installed. This solution is instead not very common in Belgium, generally replaced by a wooden flooring

system fixed to wooden beams by rivets, in the case of old masonry buildings, or composite or precast flooring system for newer reinforced concrete buildings, especially for apartment houses. These differences clearly resulted in the need for development of specific functions, more representative of the damage mechanisms (details in the Supplement). A similar process also applied to the components related to the external finishing material of the building, given the prevalence of other types of materials used in the Walloon region (e.g., stone, masonry, mixed stone-masonry), in addition to the

traditional plaster applied in Italy.

Figure 9 shows the different damage curves of INSYDE-BE and the original Italian model, obtained for a hypothetical default building with a FA=100 m$^2$ inundated by a default flood event (in this exercise, flow velocity has been set at 2 m/s in order to activate more damage components). The differences in the absolute values (Figure 9a) and shares (Figure 9b) result of all the changes reported in Table 3, with the Belgian model estimating lower losses, which cannot be totally explained by

macro-economic factors (as consumer price indices are higher in Belgium). Rather, dissimilarities between the two models mostly depend on the differences in exposure and vulnerability features as well as implemented damage mechanisms in the models for the two countries, which justifies the effort for the adaptation. In detail, Figure 9b explicitly reports the shares of the building components to the total damage computed by the two models (darker colors refer to INSYDE-BE and lighter ones to the original model) for water depths in the range 0.25-2.00 m.

**Table 3. Amendments to the original damage functions needed in INSYDE-BE**

| Damage components | | Adaptation in INSYDE-BE |
|---|---|---|
| Clean-up | Pumping | - |
| | Waste disposal | - |
| | Cleaning | - |
| | Dehumidification | Change in the fragility function depending on flood duration |
| Removal | Flooring system | New damage mechanism taking into account the different flooring system characterizing the Belgian buildings |
| | Pavement | New damage mechanism taking into account the different flooring system characterizing the Belgian buildings |
| | Baseboard | - |
| | Partition walls | Removed from the model |
| | Plasterboard | - |
| | External finishing material | New damage mechanism taking into account the different external finishing material characterizing the Belgian buildings |
| | Internal plaster | Exclusion of the contribution of the basement |
| | Doors | - |
| | Windows | Change in the fragility function depending on flood depth |
| | Boiler | Inclusion of the fragility function depending on flood depth |
| Non structural | Partitions replacement | Removed from the model |
| | Flooring system replacement | New damage mechanism taking into account the different flooring system characterizing the Belgian buildings |
| | Plasterboard replacement | - |
| Structural | Soil consolidation | - |
| | Local repair | New damage mechanism taking into account the different external finishing material characterizing the Belgian buildings |
| | Pillar repair | - |
| Finishing | External finish. mat.replacem. | New damage mechanism taking into account the different external finishing material characterizing the Belgian buildings |
| | Internal plaster replace. | Exclusion of the contribution of the basement |
| | External painting | The original function is retained, but with an additional condition depending on the external finishing material of the building |
| | Internal painting | Exclusion of the contribution of the basement |
| | Pavement replacement | New damage mechanism taking into account the different flooring system characterizing the Belgian buildings |
| | Baseboard replacement | - |
| Windows & Doors | Doors replacement | - |
| | Windows replacement | Change in the fragility function depending on flood depth |
| Building Systems | Boiler replacement | Inclusion of the fragility function depending on flood depth |
| | Radiator painting | Change in the calculation of damage extension |
| | Underfl. heating replacement | Removed from the model |
| | Electrical system replacement | Change in the calculation of damage extension |
| | Plumbing system replacement | Change in the calculation of damage extension |

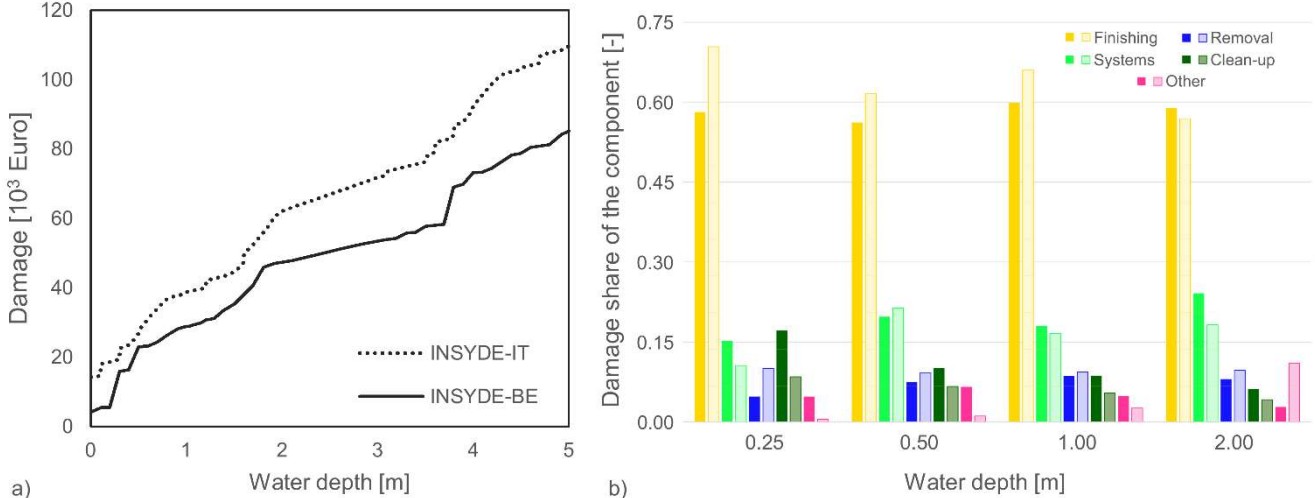

**Figure 9: a) Examples of damage functions obtained with the original Italian model and INSYDE-BE, for a default building with a FA=100 m² inundated by a default flood event (but considering a flow velocity of 2 m/s); b) Share of the different damage components to the total damage in INSYDE-BE (in darker colors) and in the Italian original model (in lighter colors).**

The figure clearly indicates the significant contribution of the damage to the finishing elements (including windows and doors) in both cases, with a share of about 60%, generally more marked for the Italian version of the model, especially at shallower water depths. Damage to building systems can be ranked as the second contributing factor, showing a share of about 15-20% and some differences between the models mainly observed for the lower and higher considered water depths. The clean-up component exhibits a peculiar pattern, being particularly important in INSYDE-BE at very shallow water depths, while in the original model it is ranked only as the fourth most influencing component, after also removal activities. The weight of residual damage components (referring mainly to non-structural elements) is observed to be larger for INSYDE-BE, except for h=2.00 m, when the corresponding share reaches about 10% for the Italian model.

## 3.2 Sensitivity analysis

Table 4 describes the baseline hazard scenarios implemented in the sensitivity analysis of damage to the hazard and building parameters. With respect to hazard parameters, four baseline scenarios were identified, as representative of typical flooding events in the Walloon region, corresponding to riverine floods, with low velocity and long duration, characterized by shallow or high water depths (scenarios 1 and 3), and flash floods with significant velocity and short duration, characterized by shallow or high water depths (scenarios 2 and 4). The reference values for the hazard parameters in the different scenarios were set according to the analysis of hazard data carried out in the development phase of the model. In detail, the reference values for the water depth for the two conditions of shallow and significant inundation were set equal to the median water depth registered for small and large rivers (0.3 m and 0.8 m) respectively; the same approach was used to set reference duration values for fast and long-lasting floods, equal to the median value registered, respectively, for all the rivers in the

Walloon region (24 hours) and for large rivers (72 hours). As regards velocity, riverine floods were characterized by the median value (0.5 m/s) of the empirical distribution identified in the analysis of the hazard maps, while flash floods by a flow velocity corresponding to the 90th percentile (1.5 m/s) of the same distribution. In the four considered baseline scenarios, the water quality was assumed to be fair ($q = 0$) while the sediment load was kept constant to the model default value (5%).

For analyzing the influence of the hazard variables, the simulations were performed assuming an attached masonry building of 75 m², with two floors and a basement, elevated 0.2 m from the ground and with medium finishing and maintenance level, as representative of a typical building of the Walloon region (as identified with the support of data statistics and surveys). Eight baseline hazard scenarios were instead identified for the sensitivity analysis of the building parameters (Scenarios 5 to 12 in Table 4). The values of the hazard features characterizing these different scenarios were selected by considering the different thresholds for these parameters causing the occurrence of the various damage mechanisms included in the model. Extensive vulnerability variables (e.g., footprint area, external perimeter, etc. and related parameters) were not included in this analysis, due to their obvious direct influence on the damage estimation. Constant parameters, or with expected small variability in practical terms (e.g., interfloor and basement height) were excluded as well.

The range of variability for each parameter in the sensitivity analysis (Table 5) was estimated by considering the maximum and minimum values observed during the elaboration of data statistics (section 3.1.1).

**Table 4: Sensitivity analysis of the model parameters: definition of the baseline scenarios for the testing of hazard and building parameters.**

| Test | Scenario | Water depth [m] | Velocity [m/s] | Flood duration [hours] | Sediment load [%] | Water quality [-] |
|---|---|---|---|---|---|---|
| Sensitivity to hazard parameters | 1 | 0.3 | 0.5 | 72 | 5 | 0 |
| | 2 | 0.3 | 1.5 | 24 | | |
| | 3 | 0.8 | 0.5 | 72 | | |
| | 4 | 0.8 | 1.5 | 24 | | |
| Sensitivity to building parameters | 5 | 0.8 | 0.5 | 10 | 5 | 0 |
| | 6 | 0.8 | 0.5 | 36 | | |
| | 7 | 0.8 | 2.5 | 10 | | |
| | 8 | 2.0 | 0.5 | 10 | | |
| | 9 | 2.0 | 0.5 | 36 | | |
| | 10 | 2.0 | 2.5 | 10 | | |
| | 11 | 0.15 | 0.5 | 10 | | |
| | 12 | 0.15 | 0.5 | 36 | | |

### 3.2.1 Results of the sensitivity analysis for the hazard parameters

The results of the sensitivity analysis are summarized in Figures 10 and 11, which show the effect (i.e., sensitivity score, Eq. 3) on the estimated damages of the different hazard (Figure 10) and building (Figure 11) parameters when they vary from the reference value established in the baseline scenarios (Table 4) to their respective extreme values (Table 5).

**Table 5: Sensitivity analysis of the model parameters: definition of the baseline values (Table 4) and lower and upper bounds for the tested parameters.**

| Parameter | Baseline values | $x_i^-$ | $x_i^+$ |
|---|---|---|---|
| Water depth $h$ [m] | 0.3 ; 0.8 | 0.3 | 3.0 |
| Flow velocity $v$ [m/s] | 0.5 ; 1.5 | 0.2 | 2.0 |
| Flood duration $d$ [hours] | 24; 72 | 12 | 168 |
| Sediment load $s$ [%] | 5 | 0 | 20 |
| Water quality $q$ [-] | 0 | 0 | 1 |
| Basement area BA [m²] | $0.5 \cdot FA$ | 0 | $FA$ |
| Ground floor level $GL$ [m] | 0.2 | 0.6 | 0 |
| Building type $BT$ [-] | 3 | 4 | 2 |
| Building structure $BS$ [-] | 2 | 1 | 2 |
| Finishing level $FL$ [-] | 1 | 0.8 | 1.2 |
| Level of maintenance $LM$ [-] | 1 | 0.9 | 1.1 |
| Year of construction $YY$ [-] | 1940 | 2000 | 1910 |
| Plant distribution $PD$ [-] | 2 | 2 | 1 |
| External finishing material $EFM$ [-] | 1 | 3 | 1 |
| Basement area BA [m²] | $0.5 \cdot FA$ | 0 | $FA$ |
| Ground floor level $GL$ [m] | 0.2 | 0.6 | 0 |

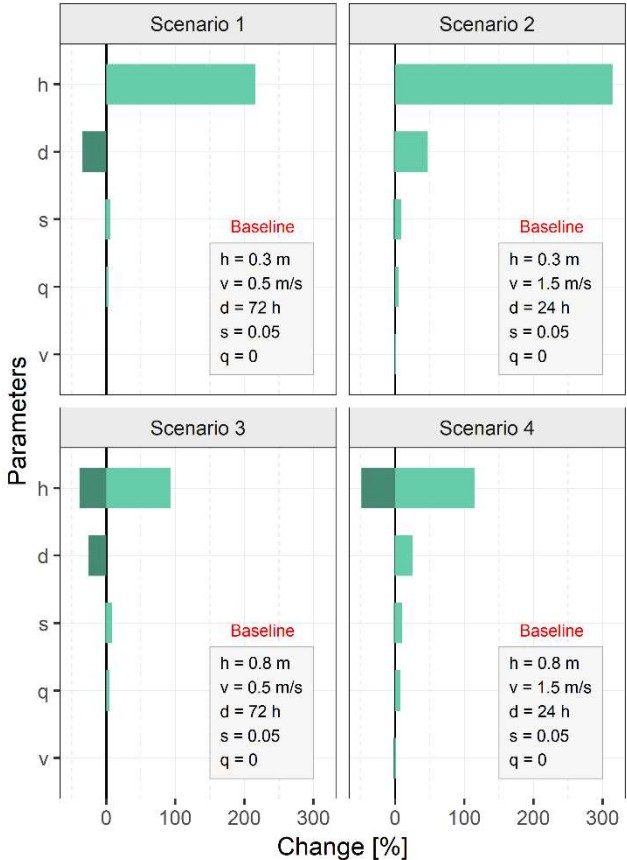

**Figure 10: Sensitivity scores for the hazard parameters (Table 5) and scenarios (Table 4). Dark green: negative change; light green: positive change.**

Consistently with the literature (Merz et al., 2013; Schröter et al., 2014; Amadio et al., 2019), Figure 10 highlights the overwhelming importance of the water depth on damage computed by INSYDE-BE, especially for shallow inundations (0.3 m, as in Scenario 1 and 2), with sensitivity scores ranging from about 200 to 300%, reducing by half in Scenarios 3 and 4, characterized by a higher reference water depth (0.8 m). A similar behavior is observed also for the flood duration, although with a comparatively smaller influence, in the range of about -35 to +45% for Scenarios 1 and 2 and ±25% for Scenarios 3 and 4.

For the case of sediment load and presence of pollutants, damage variations are found to be less significant, with sensitivity scores not exceeding 10%. Figure 10 also indicates that flow velocity is not a critical parameter for the model when assessing damages for the typical flood events occurring in the Walloon region, due to their low characteristic flow velocities (Figure 4), which cannot cause significant structural damages (i.e., a larger influence is expected instead for higher values of the flow velocity).

This analysis emphasizes the importance of the availability of high quality hazard data as input for flood damage assessment. From a practical perspective, local information on the expected water depth is in general not an issue in many countries, given that this is one of the main variables displayed in hazard maps, such as required, for example, by the European Floods Directive. The same applies for the flow velocity, which, however, in the present case, has been shown to have a negligible influence on the damage estimation. The analysis also revealed the importance of estimating the inundation duration, an information that should be derived from the application of any unsteady hydraulic model, but that is only seldom provided in the hazard maps (De Moel et al., 2009). Instead, difficulties may be encountered in assigning, e.g., in ex-ante simulations, a value to the input data related to the water quality, $s$ and $q$, which can still affect damage estimation to a certain extent (Figure 10). This problem can be overcome by applying the default values implemented in INSYDE-BE.

### 3.2.2 Results of the sensitivity analysis for the building parameters

The results of the sensitivity analysis for the building parameters (Figure 11) reveal the significant influence of the elevation of the building with respect to the ground (GL), especially for shallow water depths (Scenarios 11 and 12), with sensitivity scores exceeding 200%. This is a reasonable result, given that a certain elevation of the house can respectively avoid or reduce damage in the case of a low or a higher water depth. Other important parameters are represented by the finishing (FL) and maintenance level (LM) of the building, which induce changes in the damage estimates ranging between 15 and 40%.

The structural type of the building (BS) has a slight influence (sensitivity score < 15%) in the long-lasting inundation scenarios (6 and 9), implying an increased damage when BS = 2 (masonry), due to activation of the damage components related to the removal and replacement of the flooring system and pavement. Regarding the external finishing material (EFM), Figure 11 highlights an important effect of this parameter especially in the case of low water depth and long-lasting floods, because it affects building components like the external plaster removal and replacement, which are highly sensitive to flood duration.

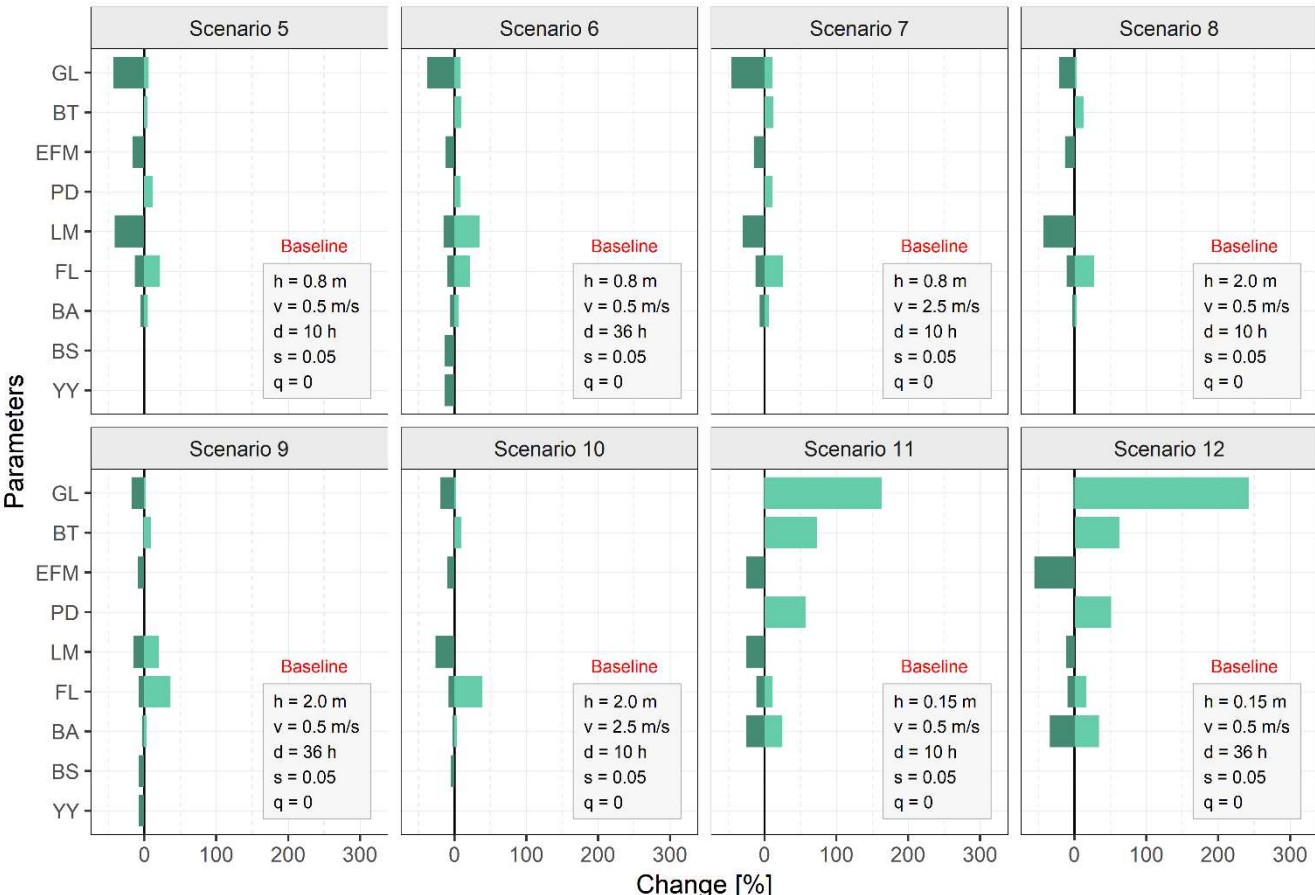

**Figure 11: Sensitivity scores for building parameters (Table 5) and scenarios (Table 4). Dark green: negative change; light green: positive change.**

The basement area (BA) is mainly important in the case of shallow inundations and, even more, when their duration is longer, as highlighted in Scenarios 11 and 12. This result can be easily explained by considering that at low water depths most of the damage components are not (or only partly) activated and then a large share of the total damage is caused by the components related to the basement. A similar pattern is found for the variable that indicates the type of distribution of the heating system (PD), due to the assumption implemented in the model regarding the location of the boiler in the building. Indeed, for the default value (PD=2, i.e., distributed system), when the water depth is shallower than 1.6 m, no damage is expected to occur to the boiler; in the case of a centralized system (PD=1), instead, the boiler is supposed to be located in the basement, which is assumed to be completely flooded for any water depth, causing then a total damage for this component. The changes of more than 50% observed for Scenarios 11 and 12 also depend on the assumption related to the costs for replacing the boiler, which are considered to be higher in the case of detached or semi-detached houses (BT=1 or 2), where more power is required.

Finally, some influence of the year of construction of the building (YY) appears only for long-lasting inundation scenarios (6 and 9 in Figure 11), although with a rather limited contribution (sensitivity score <15%), as a result of the non-occurrence of the damage to the flooring system component for newer buildings.

In conclusion, excluding the obvious contribution of the extensive geometric variables, the analysis highlighted the importance of an accurate assessment of the vulnerability parameters, especially in the case of shallow inundations, with GL

being the most critical factor, given its influence on the damage prediction and the practical difficulties for its proper characterization at the building scale. The results also indicated that the use of the default values implemented in the model can be a reasonable choice in case of lack of detailed knowledge on the other input vulnerability parameters: indeed, the analysis has demonstrated that this operation would certainly increase the uncertainty in the damage estimation, but only to a small extent.

## 3.3 Current limits to the validation of INSYDE-BE

Reliable empirical data on flood damage are essential to support the validation of flood damage models. However, such datasets remain scarce and incomplete, particularly those combining a large spatial coverage (e.g., regional, national) over a long period (e.g., several decades) with a detailed resolution (e.g., address-level data). For the Walloon region, a database of about 27,000 compensation claims submitted to the Disaster Fund (a Belgian state agency) is available over the period 1993-

2019. This database contains information on the economic damage at the building level assessed by state-designated experts for various types of natural disasters. Riverine floods correspond to about one third of the registered events between 1993 and 2019, accounting for one half of the total claimed damage. Despite this large amount of loss data, the usefulness of the Disaster Fund database for validation purposes is considered very limited, due to the nature of the compensation scheme adopted in the region in case of calamities (Doppagne, 2020; Hogge, 2020). Indeed, for inundation events, the Regional

Service of Calamities can intervene only in a suppletive way, which means that all types of damages that may be covered by an insurance contract are excluded from the compensation by the State. This supplementary character is reflected in the legislation by a limitation in the compensable assets, which mainly include only external parts of the building (a terrace on a concrete screed, a garden shed on a concrete screed, fences fixed to the ground with concrete, a retaining wall, a stone wall, etc.) and exclude the main components modelled in INSYDE. Therefore, the database provides only a portion of the overall

flood damage figure in the Walloon region, as corroborated by the detailed analysis performed by Doppagne (2020). The Author analyzed claim data related to the major flood events included in the database (1993, 1995, 2002), which caused a large portion of the total reported damage. According to the analysis, the average flood damage per building is less than 3000 €, which differs by about one order of magnitude with respect to the amounts observed in comparable flood events occurred over Europe in the last years (Thieken et al., 2005; Amadio et al., 2019). Moreover, contrary to expectations, the correlation

between claimed damage and water depth at the building scale has been found to be relatively low.

Insurance data can be certainly an alternative for validation, but their access is currently restricted due to their private nature. On the contrary, the recent floods occurred on July 2021 in Belgium (Dewals et al., 2021) can be an interesting validation test case as soon as the results of ad-hoc damage surveys in the affected areas will be made available.

In the meantime, the application of INSYDE-BE in combination with other existing models developed for contexts similar to the one under investigation can be considered to support the trust level in the damage assessment (Wagenaar et al. 2016; Figueiredo et al., 2018; Molinari et al., 2020). As an example, we report here the estimated losses obtained for a historical flood occurred between late December 1993 and early January 1994 in the lower part of the Ourthe River. For this event, it is available a validated flood extent and 5 m resolution raster of the water depths and flow velocities, simulated with the WOLF2D model (Ernst et al. 2010; Erpicum et al. 2010). The impacted areas included the villages of Tilff, Méry and Esneux, involving a total of 621 residential buildings, as extracted from the intersection of the inundated area with PICC data (Figure S11).

In this exercise, FLEMO-ps, Flemish and JRC models (Thieken et al., 2008; Vanneuville et al. 2006; Huizinga 2007; Huizinga et al. 2017) were selected as a comparison means for the results obtained by INSYDE-BE. Both the Belgian Flemish model (Vanneuville et al. 2006) and the JRC model (Huizinga 2007; Huizinga et al. 2017) use relative meso-scale depth-damage curves for damage assessment to different land-use classes. Therefore, in these two cases, average water depth and total built-up area within each inundated residential land-use zone were calculated; for each zone, the damage factor provided by the curves was then multiplied by the total exposure value of the affected buildings, based on the regional averages of housing prices. For the application of micro-scale INSYDE-BE and the German FLEMO-ps (Thieken et al., 2008), water depth and flow velocity at the building location were assigned by considering the average of raster cells within each building polygon. For INSYDE-BE, the flood duration and the parameters related to the sediment load and the presence of pollutants were set as their default values. Regarding the building features, the PICC data were used to assign the building's footprint area, while other additional vulnerability parameters were estimated using Google Street View (number of floors (NF), ground floor level (GL), building type (BT), building structure (BS) and external finishing material (EFM)) or based on the information from the census data of 2011 (year of construction (YY)). Regarding the housing quality, all the buildings were assumed to have a medium finishing level, while all other missing buildings characteristics required in INSYDE-BE were set at their default values. The results for the considered case study are reported in Table 6, which shows, similarly to other exercises on damage models' comparison (Thieken et al., 2008; Jongman et al. 2012; Scorzini and Frank 2017; Molinari et al. 2020; Paprotny et al. 2021), differences in the simulated losses up to one order of magnitude. This high spread in the results can be explained by the well-known heterogeneity characterizing the shapes of the selected damage curves, especially at shallow water depths (i.e., below 1-1.5 m; to be noted that, in the tested event, median values of water depths at building locations were about 0.3 m). An interesting observation from Table 6 is that the most similar results were provided by INSYDE-BE and the empirical model FLEMO-ps, with the last one already proven to exhibit fair performances in validation tests also in regions different from the original one (Dottori et al. 2016; Molinari et al. 2020; Paprotny et al. 2021).

**Table 6: Estimated losses for the 1993-1994 flood event of the Ourthe River (values in 2020 Euro).**

| Model | INSYDE-BE | FLEMO-ps | Flemish | JRC |
|---|---|---|---|---|
| Calculated loss [M€] | 3.1 | 2.3 | 0.6 | 10.1 |

## 4 Conclusions

In this study, we presented the adaptation of the Italian INSYDE model for the estimation of flood damage to residential buildings to the Belgian context. The procedure for the adaptation, that can be theoretically replicated in any other region

and for any other synthetic model with explicit assumptions, can be summarized as follows. After a preliminary collection of the data for the characterization of the new context in terms of hazard, exposure and vulnerability, representative values for the input parameters need to be defined by means of a statistical analysis or literature review. The next step, supported by evidence retrieved from virtual and field surveys (recommended, if possible), consists in the modification of the damage functions to represent the typical damage mechanisms occurring in the region. The updating of the unit prices for the

removal and replacement operations related to the damaged building components completes the full adaptation of the model. Nonetheless, a sensitivity analysis for the adapted model may be useful, to identify relevant variables and to analyze the effect of possible uncertainty in the input data for the damage estimation.

The study highlighted that the flexibility and the transparent methodology implemented in the INSYDE model is key for a straightforward adaptation to other contexts.

For the newly developed INSYDE-BE, the main limitation is currently related to its direct validation, which was not possible due to the lack of fully representative empirical data for the Walloon region. In this case, until useful validation data will be made available, multi-model applications, as the one shown in this study, can be useful to support a more informed damage assessments.

## Acknowledgements

The authors gratefully acknowledge the Service Public de Wallonie (SPW), and in particular SPW IAS (July 2020), for the provided data. Axelle Doppagne and Pierre Hogge are acknowledged for their contributions to preliminary analyses of damage data and model settings. Jacques Teller from the University of Liège and Francesco Ballio from Politecnico di Milano are also gratefully acknowledged for their fruitful suggestions and hints during the development of the work.

This study was partly supported by the Fonds de la Recherche Scientifique – FNRS under Grant(s) n°R.8003.18

(IC4WATER - Joint WATER JPI Call 2017).

**Authors contribution**

Conceptualization: A.R.S., D.M., B.D., Data collection: A.R.S., D.R.C., B.D., P.A., Virtual and field surveys: A.R.S., D.R.C., Data management and analysis: A.R.S., D.M., D.R.C., P.A., Investigation of results: All, Writing – original draft: A.R.S., Writing – final draft: A.R.S., D.M., B.D.

**Competing interests**

The authors declare that they have no conflict of interest.

**Code and data availability**

In order to increase the transparency and reproducibility of the methodology, the model functions are available for download as R open source code, currently hosted on Mendeley Data: doi: 10.17632/7ckzzz3xz5.1 Thus, users have the possibility of applying INSYDE-BE to compute flood damage for the different building types of interest and for any reference flood scenario. Furthermore, the model can be fully customized as users can change the value of model parameters and reference prices for the monetary evaluation of damage, as well as the different damage functions themselves. Data coming from the virtual and field surveys are available as well in the Repository (doi: 10.17632/7ckzzz3xz5.1) .

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
