# Peer review of "INSYDE-BE: Adaptation of the INSYDE model to the Walloon Region (Belgium)"

_Natural Hazards and Earth System Sciences, 2021_

## Referee Comment (RC2)

[referee-annotated manuscript omitted]

---

## Author Comment (AC2)

[revised manuscript text omitted]

---

## Author Response (AR1)

We would like to thank the Reviewers for the interest in our work and for carefully reading our manuscript; we greatly appreciate the insightful comments as they contributed to increase the manuscript robustness and, in general, to improve its quality. In the following, we provide a point-by-point reply to the general and specific comments raised.

**Reviewer 1:**

R1-C1: Transferability of damage models is a very important topic and the degree of detail and the effort put into the adaptation is impressive. However, the evaluation of the adaptation of the model is difficult. I do agree with the authors that the validation of flood damage models is usually a challenging task and hampered by missing data. Yet, I do think the evaluation and therefore also the conclusion could benefit from a few additional analyses. For instance, a comparison of the adapted and the original model. The difference between the outcomes of the models could give some indications whether the rather high effort of getting all the data sets on the building characteristics etc. is justified. Another point could be the comparison of the very detailed model INSYDE and a very simple stage damage function to assess the difference between the detailed and a more simple approach.

ANSWER: In the revised version of the manuscript, we have followed the suggestion given by both Reviewers (see also response to R2.C3) by including an additional analysis consisting in a benchmarking test for a historical flood event (Ourthe river flood occurred on December 1993-January 1994) by comparing the outcomes provided by INSYDE-BE to those of other damage models developed in the (or neighbouring) region(s) (i.e., FLEMO-ps, Flemish model and JRC). This analysis has been included in the final part of Section 3.3 (P23.L464-493 of the revised manuscript). Moreover, we have included a comparison between the new damage functions of INSYDE-BE and the ones of the original Italian model for a "default building" to better highlight how the differences in the contexts of development (mainly on the vulnerability side) justify the effort for the adaptation. This analysis has been added in the final part of Section 3.1.4 (P15.L321-344), with the inclusion of the new Figure 9.

R1.C2: Equation 3: Is i and j again the sub components of the damage? If yes, why is only i changed and j stays baseline?

ANSWER: No, in Equation 3, $i$ and $j$ do not refer to the sub-components of the damage. In this equation, notation $i$ denotes the variable under testing and $j$ refers to all the other variables that are kept constant. This issue has been clarified in the revised version of the manuscript, by including the following remark after Equation 3: "where $x_j^0$ denotes the other variables that are kept constant to their default values during the tests".

R1.C3: Also the equation only gives you one ratio, but Figure 9 and 10 are showing two outcomes for the variables (one negative and one positive). I think I get how you estimate the values for the Figures. But I think eq. 3 does not show what you actually compute. For me it seems that you compute the following:

$$(D(x_i^+, x_i^0) - D(x_i^0, x_i^0)) / D(x_i^0, x_i^0) \text{ and } (D(x_i^-, x_i^0) - D(x_i^0, x_i^0)) / D(x_i^0, x_i^0)$$

ANSWER: The Reviewer is right, thank you for pointing it out. There was a typo in the original Equation 3, that has been corrected in the revised version of the manuscript.

R1. C4: Figure 9 and 10: Please use the same x-axis in all plots to ensure a just comparison between the different plots.

ANSWER: We have made the new Figure 10 (now Figure 11 in the revised manuscript) to share the same x-axis scale as Figure 9 (now Figure 10): small positive values are now less visible than the previous version (see the following figures), but if the Reviewer and the Editor prefer this new version, it would be fine for us.

**New Figure**

[Figure]

**Original figure**

[Figure]

**Reviewer 2:**

R2.C1: I suspect a sort of circular argument underlying the sensitivity analysis. The sensitivity analysis investigates changes in the model output (damage estimate) for changes in the different model input variables (using a one at a time approach). Hence, the sensitivity of the model output to these changes depends essentially on the assumptions underlying the model structure and damage functions. Therefore you may not infer general conclusions from that, e.g. like * an 'overwhelming importance of water depth on the flood damage estimation (p17 l354)' * '… flow velocity… has been shown to have a negligible influence on the damage estimation' because this only depends on the model assumptions. Please rephrase these paragraphs in a way you have done, for instance, on p20 l 396 and rather derive some useful guidance on which data to put most effort during data collection.

ANSWER: The issue pointed out by the Reviewer is inherent to the nature of a sensitivity analysis, which is specifically aimed at investigating model's response to input data to identify the most important variables within the model. Clearly, the changes in the output depend on the assumptions underlying the model structure, but this is exactly the desired objective of a sensitivity analysis. Therefore, we would like to stress that the statements reported by the Reviewer are not "general" conclusions, but they refer to the developed model: in revising the manuscript, we have clarified this point, by specifying that the reported conclusions apply to the considered model (as in P20.L385 and P20.L391 of the revised paper).

R2.C2: Please check if equation 3 represents what you have analysed within the sensitivity analysis. Did you really consider the difference between positive and negative changes in the different inputs in relation to the reference? How do you obtain positive and negative changes from that as presented in Figures 9 and 10? It would be also good to formulate equation 3 in a way to expresses percentage changes, as it is later reported in Figures 9 and 10.

ANSWER: Equation 3 has been corrected in the revised version of the manuscript. Please see the responses C2 and C3 to Reviewer 1.

R2.C3: This entire paragraph merely tells the reader more than that no appropriate data for thorough model validation are available in the target region. Instead, it would be more interesting to present alternative approaches for model validation, e.g. picking up ideas discussed in other papers on model comparison (e.g. Wagenaar 2016, Gerl et al. 2016, or ensemble approaches Figueiredo 2018). You briefly pick up this aspect in the conclusions but I think it deserves more in-depth discussion in section 3.3. As the study does not include proper testing of the adopted model I highly recommend conducting a benchmarking exercise by comparing the outcomes of INSYDE-BE to other damage models in the region.

ANSWER: In the revised version of the manuscript, we have followed the suggestion given by the Reviewer of expanding the discussion and we have then included a benchmarking test for a historical flood event (Ourthe river flood occurred on December 1993-January 1994) by comparing the outcomes provided by INSYDE-BE to those of other damage models in the region (FLEMO-ps, Flemish model and JRC). This analysis has been included in the final part of Section 3.3 (P23.L464-493 of the revised manuscript).

R2.C4: (Section 3.1.3): Please indicate the reference year for unit prices in table S1 and indicate a resource for transferring these values to different years, e.g. statistics on building price indices.

ANSWER: The reference year for the unit prices shown in Table S1 is 2020, since we referred to the "Bordereau des Prix Unitaires 2020", as stated at P14.L293 of the original manuscript. For the sake of clarity, we have specified the reference year of the unit prices also in the caption of Table S1, where we have included indication on the possible updating of the prices by considering the construction price index (https://statbel.fgov.be/en/themes/indicators/prices/construction-output-price-index#figures).

R2.C5: 4 (Section 4): the statement that the procedure for the adoption of the INSYDE model to the Walloon region is replicable for any other synthetic model (p21 l440) is not supported by the results of this study.

ANSWER: Indeed, in the original manuscript we mentioned that the procedure can be "theoretically" applied also for other synthetic models. To be more clear, in the revised version we have specified that this could be possible for those models based on explicit assumptions on input parameters and damage mechanisms, as INSYDE (P24.L500 of the revised manuscript).

R2.C6: Please find further minor comments and typo corrections in the attached marked-up manuscript).

ANSWER: All the minor comments have been taken into consideration in the revised version of the manuscript. Please see the track-change version of the revised manuscript for details.

Additional References Wagenaar, D., Bruijn, K. M. de, Bouwer, L. M., and Moel, H. de: Uncertainty in flood damage estimates and its potential effect on investment decisions, 16, 1–14, https://doi.org/10.5194/nhess-16-1-2016, 2016.

The suggested reference has been included in the revised version of the manuscript.

---

## Editor Decision (ED1)

[revised manuscript text omitted]

---

## Author Response (AR2)

Dear Editor,

In the revised version of the manuscript, we have addressed your comments related to Figures and Tables.

A point-by-point reply to the specific comments is provided in the annotated version of the manuscript, included in the next pages of this pdf file.

Sincerely,

Anna Rita Scorzini

[revised manuscript text omitted]

---

## Editor Decision (ED2)

[revised manuscript text omitted]